# Mutational and putative neoantigen load predict clinical benefit of adoptive T cell therapy in melanoma

Martin Lauss[1], Marco Donia[2,3], Katja Harbst[1], Rikke Andersen[2,3], Shamik Mitra[1], Frida Rosengren[1], Maryem Salim[1], Johan Vallon-Christersson [1], Therese Törngren[1], Anders Kvist [1], Markus Ringnér [4], Inge Marie Svane[2,3] & Göran Jönsson[1]

Adoptive T-cell therapy (ACT) is a highly intensive immunotherapy regime that has yielded remarkable response rates and many durable responses in clinical trials in melanoma; however, 50–60% of the patients have no clinical benefit. Here, we searched for predictive biomarkers to ACT in melanoma. Whole exome- and transcriptome sequencing and neoantigen prediction were applied to pre-treatment samples from 27 patients recruited to a clinical phase I/II trial of ACT in stage IV melanoma. All patients had previously progressed on other immunotherapies. We report that clinical benefit is associated with significantly higher predicted neoantigen load. High mutation and predicted neoantigen load are significantly associated with improved progression-free and overall survival. Further, clinical benefit is associated with the expression of immune activation signatures including a high MHC-I antigen processing and presentation score. These results improve our understanding of mechanisms behind clinical benefit of ACT in melanoma.

[1] Division of Oncology and Pathology, Department of Clinical Sciences Lund, Faculty of Medicine, Lund University, Scheelegatan 2, Medicon Village, 22185 Lund, Sweden. [2] Center for Cancer Immune Therapy, Department of Hematology, Herlev Ringvej 75, 2730 Herlev, Denmark. [3] Department of Oncology, Copenhagen University Hospital Herlev, Herlev Ringvej 75, 2730 Herlev, Denmark. [4] Department of Biology, National Bioinformatics Infrastructure Sweden, Science for Life Laboratory, Lund University, Sölvegatan 35, 22362 Lund, Sweden. Martin Lauss, Marco Donia, Katja Harbst, Inge Marie Svane and Göran Jönsson contributed equally to this work. Correspondence and requests for materials should be addressed to G.Jön. (email: goran_b.jonsson@med.lu.se)

The clinical management of metastatic melanoma was revolutionized by the advent of immunotherapies. Cellular immunotherapy, namely adoptive T-cell therapy (ACT) using tumor infiltrating lymphocytes (TILs), has demonstrated very high objective response rates, and long-lasting complete tumor regression in up to 20–25% of treated patients in clinical trials[1–4]. However, ACT is a complex, costly, and highly intensive treatment which to date is reserved to patients with good performance status. About 50% of melanoma patients do not appear to have any benefit from current ACT protocols. Therefore, the development of reliable predictive criteria to identify these patients is of high clinical importance. In addition, a deeper understanding of mechanisms of primary resistance may guide modification of the classical protocols to improve the efficacy of ACT.

Intrinsic features of the TILs infused, such as the number of tumor reactive T cells, their differentiation status as well as their persistence in circulation, have been found to associate with clinical benefit from ACT[2, 3, 5, 6]. However, these analyses cannot be used to select patients to be treated as they are performed after the ACT treatment is completed.

Novel molecular analyses are providing important insights on which genomic and immunological characteristics are associated with tumor progression and response to therapy. High mutational and putative neoantigen load have been found to correlate with clinical benefit from immune checkpoint blockade therapy in lung cancer and melanoma[7–11]. Previous reports have identified neoantigens as the target of T-cell responses both in patients treated with ACT[12–14] and immune checkpoint blockade[15, 16]. Immune activation gene-expression signature defines a distinct subtype in melanoma[17, 18]. Relative prevalence of pre-existing CD8+/PD-1+/CTLA-4+ tumor infiltrating T cells appears to correlate to response to anti-PD-1[19]. Furthermore, high frequency of circulating myeloid-derived suppressor cells (MSDCs) correlates with poor response to anti-CTLA4[20, 21], whereas the expression of cytolytic markers correlates with improved response to anti-CTLA4[8].

These observations prompted us to investigate the impact of tumor molecular alterations on response to ACT in melanoma patients in a clinical phase I/II trial[6, 22]. We subjected the tumors, which TILs were derived from, to whole-exome and RNA sequencing. We show that clinical benefit is associated with high mutational and predicted neoantigen load and elevated immune signature, including a high MHC class I antigen presentation score, while absence of benefit can be linked to downregulated antigen processing and presentation machinery (APM).

## Results

**Patient cohort and tumor biopsies.** To explore the underlying biology of response to ACT in melanoma, we assembled a set of 27 patients enrolled in a clinical phase I/II trial of ACT (ClinicalTrials.gov Identifier: NCT00937625)[6, 22] and analyzed tumor samples obtained for the expansion of TILs, prior to ACT initiation. The majority of tumor biopsies were obtained from lymph node or subcutaneous metastases. All patients had previously been treated with and failed other immunotherapy, the majority receiving both IL-2 and anti-CTLA4 treatment. Notably, two patients had a mucosal primary melanoma and four had an unknown primary tumor. Two patients received BRAF inhibitors prior to biopsy and ACT, and three patients were on BRAF inhibitor treatment when biopsy was obtained for ACT. According to RECIST criteria there were five complete responders (CR), seven partial responders (PR), ten with stable disease (SD) and five with progressive disease (PD). All patients had a minimum follow-up of 37.2 months. Patients with clinical benefit

were defined as CR, PR or otherwise with an OS time of more than 2 years (Table 1). Detailed clinical characteristics of this cohort were previously reported in other publications[6, 22].

**Mutational load is associated with outcome from ACT.** The mutational landscape of 27 pre-treatment melanoma tumors and matched lymphocyte DNA was investigated by whole-exome sequencing (WES). Three out of 27 pre-treatment samples failed exome sequencing. We achieved a median coverage of 104× (median tumor coverage 98×; median normal coverage 110×). We detected a median of 286 somatic mutations in the 24 tumors (range 23–2,200). Notably, the two mucosal melanomas had 157 and 246 somatic mutations, respectively. The association of mutational burden and ACT responses is shown in Fig. 1. We found a trend ($P = 0.12$, Kruskal–Wallis test) that clinical responses defined by RECIST criteria were associated with mutational load. Patients with CR (median = 618) had the highest mutational load and patients with PD (median = 110) had the lowest mutational load while patients with PR (median = 433) and SD (median = 214) displayed intermediate mutational load (Fig. 2a). Next, we divided the cohort in patients with clinical benefit and no clinical benefit, defined as described in Methods section. We found that patients with clinical benefit harbored more somatic mutations (median = 496) as compared to those without clinical benefit (median = 169), ($P = 0.01$, Mann–Whitney test, Fig. 2b). In addition to examining mutational load in relation to response as defined by RECIST criteria and tumor regression, we also investigated mutational load in relation to patient survival. As expected, there was a significant difference in patient survival when comparing clinical benefit and no clinical benefit (Supplementary Fig. 1, $P = 2 \times 10^{-7}$, Cox regression). Patients were then stratified in three equally large groups based on mutational burden. Indeed, patients with the highest mutational burden (median 647, range 496–2200) had superior survival and patients with the fewest (median 98, range 23–193) somatic mutations had the worst outcome, while patients harboring intermediate (median 286, range 194–495) amount of somatic mutations also had intermediate survival (Fig. 2c, d). Patients belonging to the group with the highest mutational load had a 1-year progression-free survival (PFS) rate of 63%, the intermediate group 25%, and the low group had no patients with more than one-year PFS (Fig. 2c). Hence, mutational load is a strong predictive biomarker in ACT. Further confirmation that mutational load is a predictive biomarker for immune therapy came from Hugo et al.[11] demonstrating that survival after PD1-inhibition treatment is associated with mutational load. Overall, our findings suggest that mutational burden may be a good predictive biomarker for immunomodulatory agents in melanoma.

**Mutational and copy number patterns and benefit of ACT.** Several genes have been suggested to be driver genes in melanoma[18]. We addressed the role of mutations in these genes in response to ACT (Fig. 3a). BRAF V600 mutation was found in 58% and NRAS Q61 mutation in 21% of all cases. Two cases had NF1 missense mutations; however, these cases also carried a BRAF V600 and an NRAS Q61 mutation, respectively. Five cases lacked mutations in BRAF, NRAS, and NF1. Driver gene mutations were fully clonal except for a TP53 mutation in one patient (Fig. 3a). Gene mutation frequencies did not differ significantly between clinical benefit and no clinical benefit groups. Previously, mutation in the BRCA2 gene was found to be associated with intrinsic resistance to PD1 inhibition[11]. Herein, we found two cases with clinical benefit and one case without clinical benefit harboring BRCA2 missense mutation (Fig. 3a). Hence, BRCA2

**Table 1 Patient characteristics of the analyzed cohort**

| Patient ID | Previous systemic therapy[a] | Type of lesion[b] | Type of primary | RECIST | Clinical benefit | Treatment post ACT[a] | WES data | RNAseq data |
|---|---|---|---|---|---|---|---|---|
| Pat1 | IL-2 | LN | Unknown | CR | Clinical benefit | None | | Yes |
| Pat2 | IL-2, DC | LN | Skin | PD | No clinical benefit | Tem | Yes | Yes |
| Pat3 | IL-2, DC | SC | Unknown | SD | No clinical benefit | Tem | Yes | Yes |
| Pat4 | IL-2 | LN | Skin | PD | No clinical benefit | None | Yes | Yes |
| Pat5 | IL-2, Ipi, DC | LN | Skin | CR | Clinical benefit | None | Yes | Yes |
| Pat6 | IL-2, Tem, Ipi, DC | LN | Unknown | PD | No clinical benefit | None | Yes | Yes |
| Pat7 | IL-2 | LN | Skin | CR | Clinical benefit | None | Yes | Yes |
| Pat8 | IL-2, Ipi | SC | Skin | SD | Clinical benefit | Pembro | Yes | Yes |
| Pat9 | Ipi, IL-2 | LN | Skin | PR | Clinical benefit | None | Yes | Yes |
| Pat10 | IL-2, Ipi, Tem | SC | Skin | PR | Clinical benefit | ACT, Ipi | Yes | |
| Pat11 | IL-2, Ipi | NA | Skin | PR | Clinical benefit | None | Yes | Yes |
| Pat12 | Ipi, IL-2, BRAFi | LN | Skin | PR | Clinical benefit | None | Yes | |
| Pat13 | IL-2, Ipi | LN | Skin | SD | No clinical benefit | BRAFi | Yes | Yes |
| Pat14 | Ipi | IM | Skin | CR | Clinical benefit | None | Yes | Yes |
| Pat15 | Ipi, IL-2, BRAFi | SC | Skin | PD | No clinical benefit | None | Yes | Yes |
| Pat16 | IL-2, DC, Ipi, BRAFi | LN | Skin | SD | No clinical benefit | Tem | | Yes |
| Pat17 | IL-2, Ipi | SC | Mucosal | PR | Clinical benefit | Tem | Yes | Yes |
| Pat18 | IL-2, Ipi, Tem | LN | Mucosal | SD | No clinical benefit | Nivo | Yes | Yes |
| Pat19 | IL-2, Ipi, BRAFi | SC | Skin | SD | No clinical benefit | Pembro, BRAFi, BRAFi/MEKi, Tem | Yes | Yes |
| Pat20 | IL-2, Ipi, BRAFi | IA | Skin | PR | Clinical benefit | ACT, Pembro, BRAFi, Tem, Sel, KPT-330 | Yes | Yes |
| Pat21 | IL-2, Ipi | SC | Skin | SD | No clinical benefit | BRAFi, Tem | Yes | Yes |
| Pat22 | Ipi, IL-2 | IA | Skin | SD | No clinical benefit | BRAFi, Tem, Pembro, | Yes | Yes |
| Pat23 | IL-2, Ipi | SC | Skin | CR | Clinical benefit | None | Yes | Yes |
| Pat24 | IL-2, Tem | SC | Unknown | PD | No clinical benefit | Ipi, Pembro | Yes | Yes |
| Pat25 | IL-2, Ipi, BRAFi | LN | Unknown | PR | Clinical benefit | Pembro | | Yes |
| Pat26 | IL-2, Ipi | LN | Skin | SD | Clinical benefit | Pembro, BRAFi, BRAFi/MEKi, Tem | Yes | Yes |
| Pat27 | IL-2, Ipi | Pleura | Unknown | SD | No clinical benefit | Pembro, Tem | Yes | Yes |

[a]BRAFi BRAF inhibitor; DC Dacarbazine; Ipi Ipilimumab; MEKi MEK inhibitor; Nivo Nivolumab; Pembro Pembroluzimab; Sel Selinexor; Tem Temodal
[b]IA intra-abdominal; IM intra-muscular; LN lymph node; SC subcutaneous

mutation cannot explain poor response to ACT in our cohort. While UV-radiation induced DNA damage was the dominant source of mutations, the presence of additional mutational signatures was suggested in a few patients (Supplementary Fig. 2).

We next investigated DNA copy number changes in relation to response to ACT. The fraction of genome altered was not associated with clinical benefit after therapy (Fig. 3b). Of the genes with known amplifications and deletions in melanoma, the *CDKN2A* locus showed a considerable deletion frequency (Fig. 3a). However, the cases with *CDKN2A* loss had diverging RECIST outcome. Notably, deletions of the interferon (IFN) locus located next to the *CDKN2A* gene have been shown to associate with resistance to anti-CTLA4 treatment[23]. In this cohort, five tumors in the clinical benefit group and three tumors in the no clinical benefit group harbored deletions of the IFN locus, hence there was no statistical difference between the groups ($P > 0.6$, Fisher's Exact test), Acquired loss-of-function alterations in the *B2M* gene, whose protein product beta-2-microglobulin is an essential part of the MHC-I complex, has been identified in relapse lesions following treatment with PD1 inhibitor[24]. In our

cohort, the *B2M* gene was heterozygously deleted in six patients and four of these patients had a PFS <1 year (Fig. 3a). Three of the six patients with *B2M* deletion had a clinical benefit from ACT and two of these were objective responders according to RECIST (Fig. 3a). Furthermore, no difference in *B2M* gene expression was found between cases with or without *B2M* loss ($P = 0.30$, $t$-test) suggesting that *B2M* is still functional in deleted cases. Importantly, no case harbored *B2M* somatic mutation. Moreover, no difference in DNA copy number gains at the HLA-locus was identified and we only found one case of *PD-L1* gene amplification (Fig. 3a). Collectively, specific mutations or DNA copy number alterations are not predictive of response to ACT.

**Immune activation in patients who benefit from ACT.** Previous studies have defined transcriptional melanoma subgroups expressing high levels of immune response associated genes[17, 18]. Increased messenger RNA levels of such genes are associated with an improved outcome in primary and stage III metastatic melanoma[25, 26]. Thus, it is tempting to speculate that such tumors may

have an improved response to immunotherapy. Herein, we performed RNA sequencing of 25 melanoma biopsies of the 27 samples obtained prior to ACT to determine transcriptional features associated with response to ACT. First, we used the 1500 most variable genes and performed hierarchical clustering and did not observe any clear correlation between response to ACT and the cluster dendrogram (Supplementary Fig. 3). Next, we classified all tumors using the IPRES signature previously shown to be predictive of response to PD1-inhibition[11]. In total, we found five IPRES-enriched cases in patients with no clinical benefit and five IPRES-enriched cases without clinical benefit ($P = 1$, Fisher's Exact test). OS rates were comparable for IPRES-enriched and IPRES-not-enriched cases ($P = 0.8$, Cox regression, Supplementary Fig. 4) thus suggesting that the IPRES signature is not predictive of response to ACT.

We then compared the transcriptomes of tumors from patients displaying clinical benefit to those with no clinical benefit. Gene set enrichment analysis (GSEA) indicated the up-regulation of immune system associated genes in patients with clinical benefit, and in addition indicated a role for IFN-gamma signaling (GSEA, fdr = 0). This was supported by gene ontology analysis using DAVID[27] (Supplementary Table 1). Tumors from patients with no clinical benefit were found to have relatively high expression levels of genes involved in the cell cycle (GSEA, fdr = 0, Fig. 4a). Although, significant pathways in the GSEA included chromosome maintenance (fdr = 0) and meiotic recombination (fdr = 0) it was dominated by cell cycle and proliferation related signatures (Supplementary Table 1). Further analysis showed that the core genes of the MHC-I antigen presentation pathway were strongly correlated across the cohort and that the expression of several antigen presentation genes was relatively high in tumors from patients with clinical benefit. Thus, we constructed a gene-expression score for the activity of MHC-I antigen presentation

(Fig. 4b, Supplementary Fig. 5) and observed that the samples with the highest MHC-I score (top 25%) were all derived from patients with excellent outcome albeit the PFS analysis was not significant (Fig. 4c). These results suggest that up-regulation of immune system associated genes and MHC class I dependent antigen presentation is associated with ACT efficiency in melanoma. To further dissect these findings, we again turned to the Hugo et al. dataset but intriguingly could not find a strong correlation of MHC-I antigen presentation genes (Supplementary Fig. 6). Furthermore, we did not observe a difference in survival based on MHC-I gene score activity in the anti-PD1 treated cohort (Supplementary Fig. 6). Finally, we analyzed the prognostic effect of the MHC-I gene score in the TCGA[18] and Cirenajwis et al.[26] cohorts. In both cohorts the MHC-I genes were tightly co-expressed and showed a prognostic effect in metastatic melanoma (Supplementary Fig. 6).

Recently, Tirosh et al.[28] performed single-cell RNA sequencing of cancer and immune cells derived from metastatic melanoma tumors and generated single immune cell signatures. In order to elucidate the composition of the immune cell infiltrate and potential role of individual immune cell types in response to ACT in melanoma, we applied these signatures to our data. For further comparison we applied these signatures to the TCGA data as well. Here, we found T- and B-cell signatures associated to survival in stage IV melanomas (Supplementary Fig. 7). However, when applying the signatures to the ACT data we found no association between any particular immune cell type and clinical benefit (Fig. 4d) and furthermore no association to survival was observed (Supplementary Fig. 8). Similarly, we found no association of single immune cell markers (Fig. 4d) or ratios thereof (Supplementary Fig. 9) to clinical benefit. These analyses indicate that immune cell infiltration was favorable for ACT outcome, rather than the presence of a specific immune cell type.

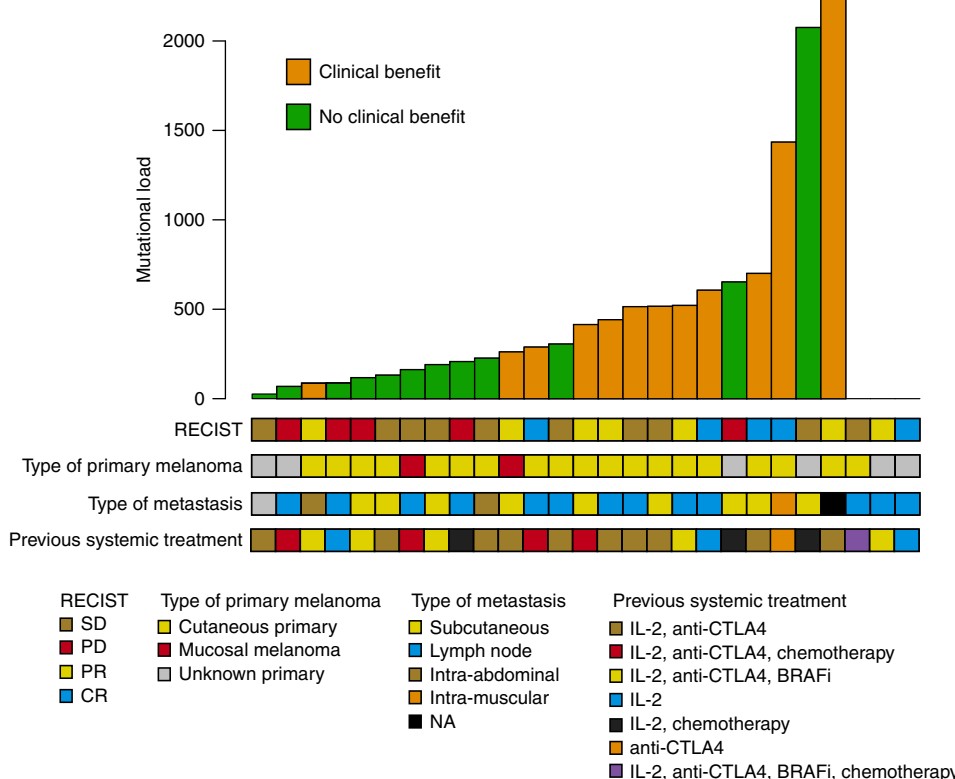

**Fig. 1** Mutational load and clinical features of the cohort. Tumors are ordered according to mutational load and compared to clinical features. RECIST categories: CR complete response; PD progressive disease; PR partial response; SD\ stable disease

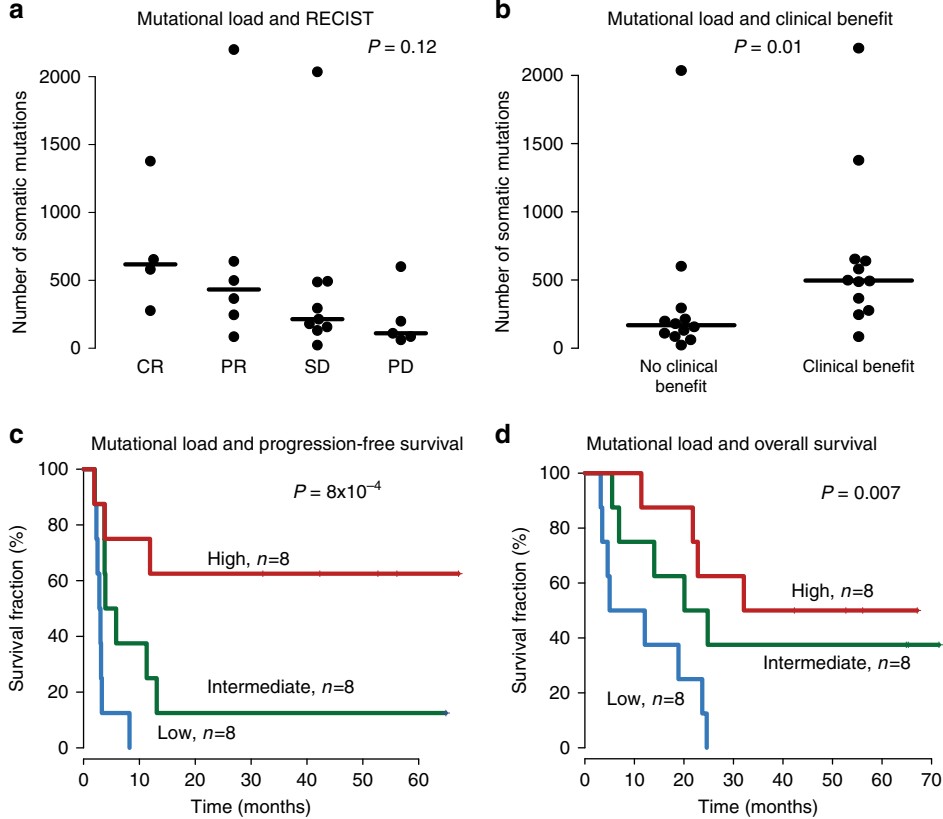

**Fig. 2** Mutational load is associated with benefit from ACT in melanoma. **a** Tumor mutational load by clinical response according to RECIST. *P*-value from Kruskal–Wallis test. Solid black line indicates the median. **b** Mutational load by clinical benefit groups, as defined in Methods section. *P*-value from Mann–Whitney test. **c** Tumor mutational load is significantly associated with progression-free survival, when the cohort in divided into three groups by mutational load. *P*-value from Cox regression. **d** Tumor mutational load is significantly associated with overall survival, when the cohort in divided into three groups by mutational load. *P*-value from Cox regression. RECIST categories: CR complete response; PD progressive disease; PR partial response; SD stable disease

Intriguingly, both T-cell exhaustion markers and IFN-signaling genes showed a trend towards up-regulation in tumors from patients with clinical benefit, with the majority of the genes not reaching significance. Genes demonstrating some association to clinical benefit included *CTLA4*, *HLA-C*, and *TAP2* ($P < 0.05$, *t*-test) however there were several genes displaying borderline association including *JAK2*, *PDL1*, *HLA-B*, *HLA-DRB1*, and *B2M* (Fig. 4d). The role of these pathways that control T-cell activity will have to be explored in a larger dataset. Melanoma cell lineage signatures, defined by MITF and AXL scores[28], as well as cAMP-signaling associated genes, previously linked to MAPKi resistance in melanoma[29], were equally expressed in tumors from patients with and without clinical benefit (Fig. 4d), suggesting that the activity of melanoma cell intrinsic programs does not influence clinical efficacy of ACT. Finally, we wanted to explore the relationship between tumor-immune gene-expression signatures and mutational load. We used the T-, B-cell and macrophage signatures by Tirosh et al[28] however we did not find any correlation between expression signatures and mutational load (Supplementary Fig. 10). Consequently, no correlation was found between the MHC-I gene-expression score and mutational load (Supplementary Fig. 10). Collectively, tumor-immune micro-environmental processes rather than melanoma lineage transcriptomic signatures are associated with ACT efficacy in melanoma. Such tumor-immune signatures are independent of mutational load.

**Predicted neoantigen load is associated with benefit of ACT.** Melanoma is one of the most neoantigen rich cancer forms[30]. The

association of high predicted neoantigen load and favorable outcome following immune checkpoint inhibition therapy has been reported in NSCLC and melanoma[8–10]. We sought to address the relationship between putative neoantigen load and clinical benefit from ACT in patients who previously progressed on immune therapy. We identified a median of 96 predicted neoantigens in the 23 tumors (range 6–709). Of these, 31 (37%) were expressed (range 4–183) as assessed by variant allele read counts from RNA sequencing data (Fig. 5a). Interestingly, *BRAF* V600E mutation was predicted to form a neoantigen when presented by HLA-A0301 or HLA-A1101. Similarly, *NRAS* Q61K/L/R mutations were predicted to generate neoantigens in the presence of certain HLA-A alleles. We assessed whether putative neoantigen load was associated with clinical benefit in our study. When dividing the cohort by RECIST-defined response, median predicted neoantigen count followed response, with patients achieving CR having a median of 85 expressed predicted neoantigens, PR–32, SD–20, and PD only 13 predicted neoantigens (Fig. 5b). Furthermore, tumors obtained from patients with clinical benefit harbored a median of 58 expressed predicted neoantigens, while those with no benefit only had 18 (Fig. 5c). Finally, we assessed whether tumor neoantigen load correlated with patient outcome. Patients with the lowest predicted neoantigen load (median 8, range 4–18) had worst survival as compared to patients with the highest (median 85, range 57–183) predicted neoantigen load and intermediate (median 31, range 19–56) amount of predicted neoantigens (Fig. 5d). Importantly, predicted expressed neoantigen burden correlated strongly with mutational load (Pearson correlation 0.98). In conclusion, predicted

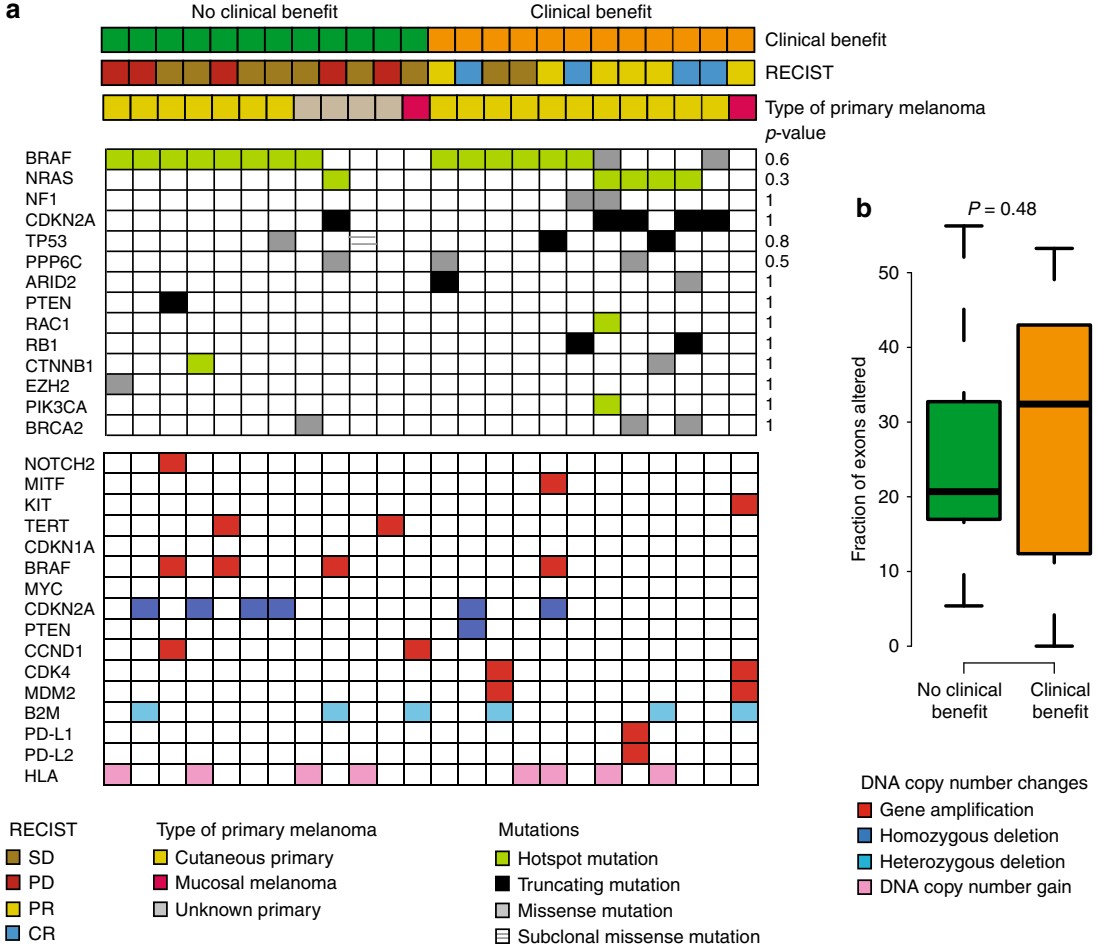

**Fig. 3** Driver gene mutations and copy number aberrations are not associated with clinical benefit of ACT in melanoma **a**. Mutations in melanoma driver genes in tumors from patients without (left) and with (right) clinical benefit from ACT. There is no significant difference in mutation frequency between the groups in any of the genes, as shown on the right by the *P*-values from logistic regression with adjustment for mutational load of samples. **b** Fraction of exons with copy number changes for tumors from patients with and without clinical benefit. *P*-value from Mann–Whitney test. In this standard boxplot, the center line represents the median, the box limits represent the lower and upper quartiles, the whiskers extend to the most extreme values within 1.5xIQR. RECIST categories: CR complete response; PD progressive disease; PR partial response; SD stable disease

neoantigen load is associated with clinical benefit from ACT in melanoma.

To test the independent effect of predicted neoantigen load and the MHC-I score we used a multivariate Cox regression model. Using log10-transformed predicted neoantigen count and MHC-I gene-expression score values as variables, both were found to be independent predictors of OS ($P = 0.007$ and $P = 0.01$, respectively, Cox regression). Collectively, these results demonstrate that predicted neoantigen load is associated with OS and is independent of tumor-immune micro-environmental gene-expression signatures.

## Discussion

The high response rates with long-term complete tumor regressions make ACT a highly promising therapeutic modality in metastatic melanoma. Currently, ACT is being developed in several other forms of solid tumors including sarcomas[31], cervical-[32], ovarian-[33], renal-[34] and gastrointestinal cancers[35]. In an attempt to increase the efficacy of ACT, various adjustments to the classical treatment protocol and combinations with other therapeutics are under investigation in multiple trials (source: clinicaltrials.gov). However, although there is an unmet need for patient stratification prior to ACT, patients with melanoma are currently not preselected for ACT clinical trials based on tumor characteristics, since these have not been comprehensively studied in relation to response to ACT.

Here, we report the results from a comprehensive genomic analysis of tumor samples from a phase I/II clinical trial of ACT in melanoma. Our data provide compelling evidence that a high mutational and predicted neoantigen tumor load is associated with improved clinical outcome following ACT. The same phenomenon was previously reported to associate with the outcome of patients with melanoma or lung cancer, following treatment with immune checkpoints inhibitors[7–10], highlighting the importance of neoantigens in response to immunotherapy. Since melanoma and lung cancer have the highest average mutational load of all tumor types[36], this phenomenon probably reflects an increased likelihood of forming neoantigens that will elicit T-cell reactivity, thereby explaining why unselected melanomas also show the highest clinical response rates following current checkpoint-immunotherapy[37]. Importantly, all patients enrolled in the current study had failed on prior immunotherapies such as intravenous IL-2 and/or anti CTLA-4 antibodies, and mutational load still was significantly associated with clinical benefit from ACT. Although, these results are intriguing larger studies are a necessity to refine thresholds of mutational load as well as further validation of mutational load as

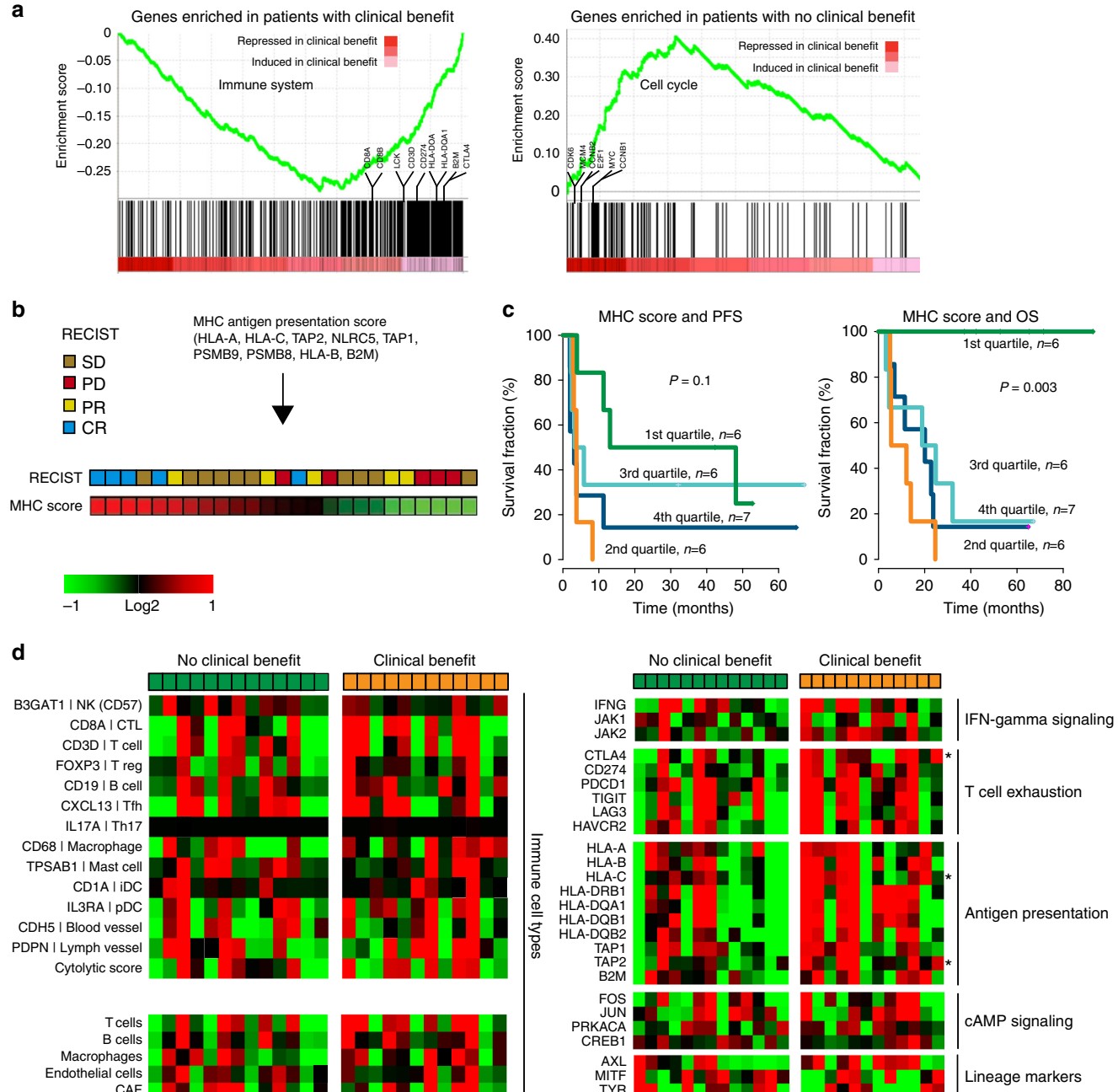

**Fig. 4** Expression of immune related genes and association to benefit from ACT in melanoma. **a** Enrichment plots from gene set enrichment analysis for the immune system and cell cycle pathways. **b** MHC class I expression score was constructed from indicated genes. **c** Progression-free and overall survival by MHC score quartile. *P*-values from Cox regression. **d** Expression of immune response related and melanoma lineage genes in patients with no clinical benefit (green) and with clinical benefit (orange). Signatures in the lower left panel (T and B cells, Macrophages, Endothelial cells, and CAF) are derived from publication by Tirosh et al.[28]. Star denotes genes with significant association to benefit from ACT (*P* < 0.05, *t*-test)

a predictive biomarker. In addition, in this study we analyzed the same biopsies used to manufacture TILs for infusion. In order to fit into common clinical settings, it might be desirable to know in advance the likelihood of a given individual patient to benefit from ACT. Thus further validation of these biomarkers with genomic analyses performed on earlier tumor biopsies, e.g., obtained before enrollment in ACT protocol, is needed. Importantly, the potential use of such predictive biomarkers would fit into current clinical paradigms because, after the introduction of checkpoint- and BRAF- inhibitors, ACT is no longer considered as a first line therapy in melanoma because of

toxicity and complexity. Therefore, performing such genomic analyses early in the metastatic disease may leave ample time to identify patients that potentially will benefit from ACT as salvage therapy.

Presentation of tumor-antigens through the MHC class I APM pathway is required for tumor-recognition from cytotoxic CD8 + T cells[38]. Previous case reports have shown acquired loss of MHC class I antigen presentation, e.g., through loss of B2M in melanoma metastases, suggesting importance of T-cell mediated immune response at advanced stages of disease[39, 40]. Here, a high expression of MHC class I APM genes was associated to clinical

benefit. Four out of six patients with a heterozygous loss of the *B2M* gene had a very poor PFS; however three patients with *B2M* loss had a clinical benefit of ACT. This indicates that there may be *B2M*-independent mechanisms for T-cells to recognize the tumor cells and that *B2M* loss alone is not a good predictive marker for ACT response. Intriguingly, analyses of previously published patient cohorts revealed that MHC class I APM pathway genes were not co-expressed, and did not predict response to PD1 inhibition in melanoma[11]. While all patients in the present study had received and failed immunotherapies prior to ACT, there is no record of previous immunotherapy in the cohort from Hugo et al.[11]. This observation might reflect immunological editing of tumors induced by prior immunotherapies in some patients. Taken together, these findings suggest that functioning of the MHC class I APM above a certain threshold level is a necessary condition to respond to immunotherapies based on T-cell attack. One important implication is that loss of an efficient MHC class I APM may prevent the

beneficial effects of the same immunotherapies, but further studies should clarify whether this can be used as an entry criterion for ACT protocols and/or other immunotherapies.

In this study, although general activation of the immune response was significantly associated with clinical benefit from ACT, we did not observe association of specific immune cell type markers (e.g., T cell, B cell, macrophage, and regulatory T cell) or various ratios thereof, including cytolytic activity[41], to clinical benefit. Taking into account previous reports showing that T-cell infiltration[17] and T-cell expression signature[42] are positive prognostic factors in immunotherapy naïve patients, as we also demonstrate in the TCGA cohort, this may reflect the effect of prior immunotherapies on immune cell infiltration, as these signatures were not predictive of survival in the current cohort of patients treated with ACT. While loss of MITF is associated to resistance to MAPK inhibition[43], expression of *MITF* and *AXL* was not associated to clinical benefit from ACT in our data; therefore, melanoma cell features such as lineage and

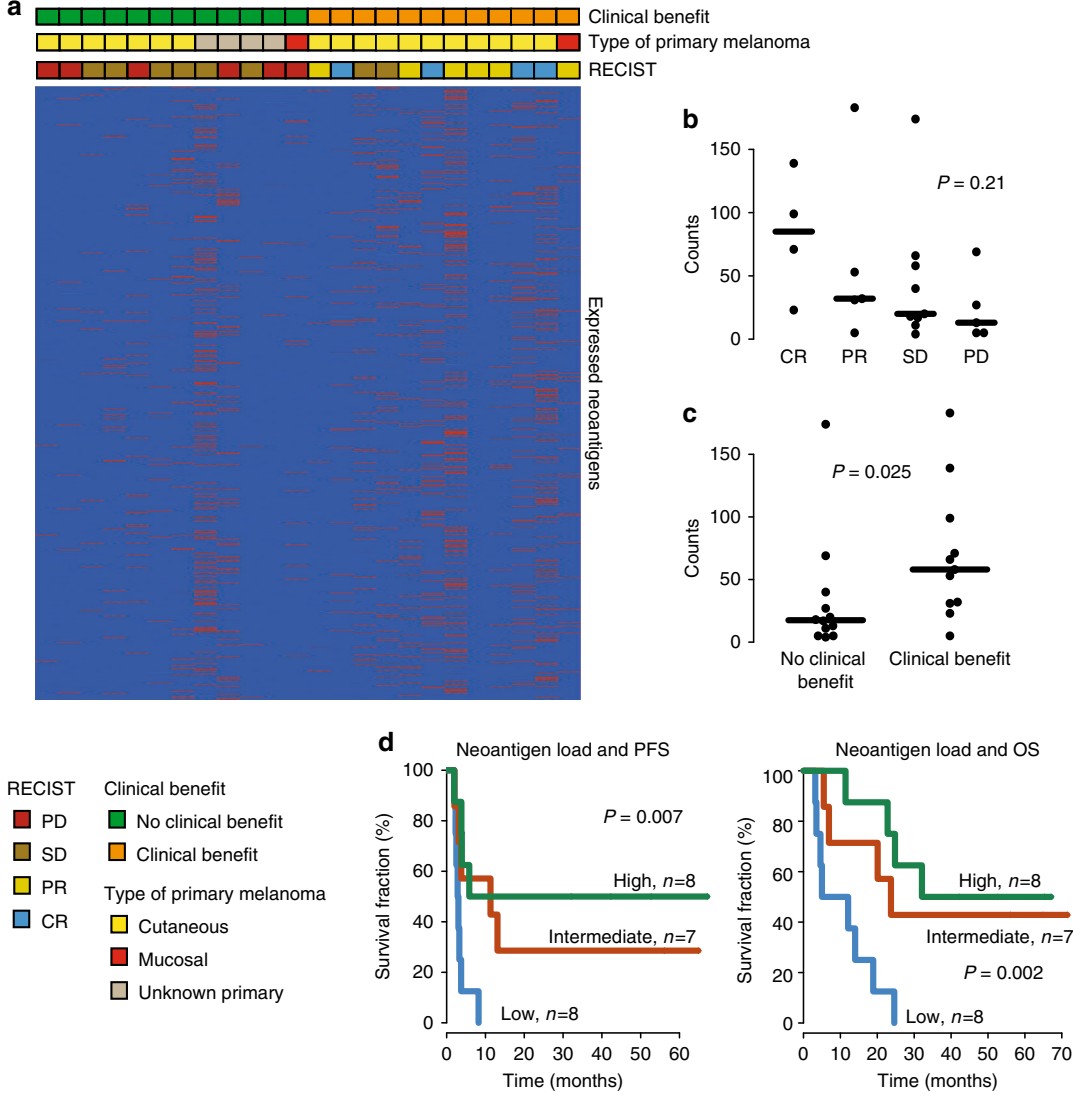

**Fig. 5** Neoantigen load is associated with benefit from ACT in melanoma. **a** Expressed predicted neoantigens are depicted in patients with and without clinical benefit from ACT. **b**, **c** The number of expressed predicted neoantigens by RECIST response **b** and clinical benefit status **c**. *P*-value from Kruskal–Wallis and Mann–Whitney tests, respectively. Solid black line indicates the median. **d** Expressed predicted neoantigen load is significantly associated with progression-free (PFS) and overall survival (OS). *P*-value from Cox regression. RECIST categories: *CR* complete response, *PD* progressive disease, *PR* partial response, *SD* stable disease

differentiation may play a minor role in defining response to immunotherapy. Finally, up-regulation of cell cycle genes appears to be an adverse predictive factor of ACT in melanoma, in line with previous findings of poor prognosis associated with proliferation in melanoma[17, 25, 26]. Intriguingly, our results further demonstrated that predicted neoantigen load and tumor-immune gene-expression signatures are independent predictors of survival in this ACT treated cohort. This suggests that a composite biomarker of neoantigen load and immune signatures should be explored for prediction of response to immunotherapy in melanoma.

In conclusion, our findings reveal tumor molecular features associated with response to ACT that warrant further investigation. If confirmed by independent studies, they can be used to guide the application of ACT in melanoma and, potentially, other solid tumors where ACT is currently under development.

## Methods

**Patient cohort and material**. All patients of this study were enrolled in the clinical trial NCT00937625[6, 22], where the efficacy of ACT followed by attenuated doses of interleukin-2 was investigated. All patients signed a written consent form. All patients had stage IV melanoma and had received prior systemic therapy. Response was assessed according to RECIST 1.0. In the cohort of 27 patients enrolled in the trial the median overall survival (OS) was 22.8 months. Further, for the purpose of statistical analyses, the cohort was divided into clinical benefit and no clinical benefit; all patients with tumor response according to RECIST (PR or CR) or OS of at least 2 years were defined as patients with clinical benefit. Two years was selected as cutoff because of the median OS of 22.8 months. Only two patients in the clinical benefit group had an OS of less than 2 years, however both patients had an objective tumor response according to RECIST. Clinical data are summarized in Table 1. Altogether, pre-treatment tumor samples from 27 patients could be analyzed. Following data were missing from pre-treatment samples: gene-expression data: patients 10 and 12; copy number and mutation data: patients 1, 16, and 25. For survival analysis, the database was locked on June 27th 2017 (Supplementary Data 1).

**Nucleic acid extraction and sequencing**. Tumor DNA and RNA were extracted using AllPrep DNA/RNA Mini Kit (Qiagen) from snap frozen or frozen in DMSO tumor fragments. Normal DNA was derived from PBMCs or TILs using QIAamp DNA Mini Kit (Qiagen). Tumor and normal DNA was subjected to WES library preparation using SureSelect Target Enrichment System for Illumina Paired-End Sequencing Library Protocol (Agilent Technologies) with Clinical Research Exome (CRE) capture oligo panel (Agilent Technologies). Barcoded WES libraries were pooled and sequenced on HiSeq 2500 (Illumina) in paired-end mode. RNA sequencing was performed as previously described[44].

**WES data analysis**. WES analysis including alignment, post-alignment processing, and variant calling from WES data was performed as described previously[44]. Briefly, reads were mapped to the human reference genome (hg19) with decoy using Novoalign (Novocraft Technologies), and duplicate fragments were marked using the MarkDuplicates functionality of Picard tools. Local realignment and base quality score recalibration were done using GATK. For variant calling, MuTect[45] version 1.1.4 (default settings) and VarScan[46] version 2.2.8 were used; for VarScan somatic, minimum variant allele frequency in the tumor was set to 10%; following recommendations from VarScan developers, somatic single-nucleotide variants (SNV) calls were further filtered to remove potential false positives; finally only high confidence calls were retained. For SNV, consensus of the two callers was retained, while indels were derived using VarScan only. For variant annotation and translation into protein sequence, Annovar[47] was used; only mutations within the coding sequence (CDS) regions of the genes were retained. Mutation data can be found as Supplementary Data 3. Statistical analyses of total mutational load are based on all mutations found in a particular sample. DNA copy number data were generated using Contra[48] version 2.0.3, and data were segmented using GLAD[49]. DNA copy number data derived from WES data can be found as Supplementary Data 2. Subclonality of mutations in the form of 'cancer cell fraction' was obtained using ABSOLUTE[50] version 1.0.6, with the setting copy_num_type = total and without a minimum mutation allele frequency; the top model was selected. Mutational signatures were derived using the R package *deconstructSigs*[51] with the reference signatures *signatures.cosmic*[36].

**HLA typing and neoantigen prediction**. For HLA typing, sequencing libraries were prepared from normal DNA using Illumina TruSight HLA Sequencing Panel according to manufacturer's instructions (Illumina) and sequenced on a MiSeq instrument (Illumina). In addition, HLA type was derived from WES data of normal (non-tumor) samples using bwakit (https://github.com/lh3/bwa/tree/

master/bwakit) or Omixon Target HLA (Omixon). Potential HLA class I restricted neoantigens were predicted using a custom pipeline comprising modified pVAC-Seq[52]. This early perl based version of pVAC-Seq was not able to generate predictions for indels. NetMHC version 4.0 was used for peptide-MHC affinity predictions; mutated peptides with binding affinity below 500 nM were retained; pVAC-Seq sequencing depth based filter was not applied. Neoantigens supported by minimum 2 variant RNA reads were considered expressed. In this study, we have only used MHC class I predictions.

**Expression signatures**. RNAseq data were processed as previously described[53] using TopHat2[54] and Cufflinks[55] v2.1.1. Isoform FPKMs were summed up to obtain gene-level expression. Data were quantile-normalized and log-transformed by $\log_2(\text{data} + 1)$. Genes were median-centered and reduced to protein-coding genes defined by the HGNC. SAM analysis[56] was used to rank genes based on differential expression scores, DAVID[27] was used for GO-term analysis, and GSEA with C2, C6, and C7 gene lists was used for GSEA[57]. Single-cell signatures and melanoma lineage signatures were from Tirosh et al.[28]. The MHC-I APM displayed high correlation of gene expression. In particular, HLA-A, HLA-B, HLA-C, TAP1, TAP2, NLRC5, PSMB9, PSMB8, and B2M were highly correlated, and further termed the "core" MHC-I set (Supplementary Fig. 5). The mean expression of these core genes is the MHC-I score and was then divided in quartiles across patients to test the association with OS. For the MHC-I score quartiles, the resulting bins were: 4th quartile [−2.98,−0.493], 3rd quartile (−0.493,0.196], 2nd quartile (0.196,0.728], and 1st quartile (0.728,2.16]. The MHC-I score was subsequently applied to the TCGA[18] and Cirenajwis et al. cohorts[26]. To stratify the patients we again used the MHC-I score quartiles determined in the respective data sets. We applied the IPRES signatures to our cohort as specified in Hugo et al.[11]. Briefly, 21 of 22 validated IPRES signatures were available, whereof 15 gene sets were available from Broad MSigDB (http://software.broadinstitute.org/gsea/msigdb/) and six gene sets were available from Supplementary Data from Hugo et al.[11]. The gene set variation analysis (GSVA) scores were calculated from uncentered gene-expression data; GSVA scores were transformed to z-scores and the mean z-score of the signatures was obtained[58]. A cutoff of >0.35 was applied as described in Hugo et al.[11] for a sample to be called "IPRES-enriched".

**Data availability**. RNAseq data have been deposited at Gene Expression Omnibus with accession number GSE100797. Clinical annotation data, DNA copy number data, and somatically called mutations are available as Supplementary Data 1–3, respectively. All other remaining data are available within the article and Supplementary Information Files, or available from the authors upon request.

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

## Acknowledgements

The study was supported by the Swedish Cancer Society, the Swedish Research Council, BioCARE, the Berta Kamprad Foundation, the King Gustaf V Jubilee foundation, the Gunnar Nilsson Cancer foundation, Mats Paulsson's foundation, Stefan Paulsson's foundation, and the governmental funding for healthcare research (ALF). The Danish Cancer Society, the Aase and Einar Danielsen's Fund and the Capital Region of Denmark Research Foundation. The research leading to these results has received funding from the European Community's Horizon 2020 Framework Programme for Research and Innovation (H2020-MSCA-ITN-2014) under Grant Agreement no. 247634. This project has received funding from the European Union's Horizon 2020 research and innovation programme under the Marie Skłodowska-Curie grant agreement No 641458.

## Author contributions

M.L., M.D., K.H., I.M.S., and G.J. conceived the study and drafted the manuscript. K.H., F.R., and T.T. performed laboratory analyses. M.L., K.H., A.K., J.V.C., S.M., M.S., and M.R. performed bioinformatic analyses. M.D., R.A., and I.M.S. collected clinical information. M.L., M.D., K.H., R.A., S.M., F.R., M.S., J.V.C., T.T., A.K., M.R., I.M.S., and G.J. read and approved the final manuscript.
