## [Peer Review File · Nature Communications]

Reviewers' comments:

Reviewer #1 (Remarks to the Author):

Mutational and neoantigen load predict clinical benefit of adoptive T cell therapy in melanoma

The major claims of this manuscript are that a high rate of non-synonymous mutations and higher predicted neoantigen load are associated with clinical benefit in patients with metastatic melanoma treated with adoptive transfer with TIL. In addition, a strong immune activation signature is also associated with clinical benefit.

Given the current need for predictive biomarkers for immunotherapy of all kinds, the research questions addressed in this manuscript are vital to ongoing clinical efforts of adoptive cell transfer.

Points to be addressed:

It is an ongoing challenge in this field that "predicted neoantigens" and "neoantigens" continue to be used interchangeably, with few studies experimentally validating in silico predictions with in vitro recognition.

When defining clinical benefit as a means of grouping patients, it is necessary to know if the patients received any therapy after adoptive cell transfer. It would also be of interest to know why a time period of 2 years was selected as a cut-off, as the majority of studies that have begun to use this criteria (checkpoint blockade) often use 6 months. Additionally, from the overall survival curves provided (Supp Fig 1), it appears that perhaps 3 or 4 responding patients did not survive 24 months (three steps and a censor mark?) which doesn't account for what may be biological differences between a patient with a >2y stable disease and a patient with a short-lived PR with limited survival.

The patients were stratified into three groups for mutational burden, but it is unclear how those groups were chosen. Was it driven by median values of the experimental group? Split into three equally populated groups? Or binned by discretely chosen values? It would be helpful to know more not only about the grouping, but about the filtering criteria applied to the samples in defining non-synonymous mutations. If this assessment of mutational burden is to be validated by other groups as a potential biomarker, it needs to be reproducible.

The expression data is intriguing and nicely presented but would be enhanced by an explanation of the derivation of the MHC gene-expression score. Also, the TCGA and Cirenajwis cohorts were described as a validation set for this score, with a significant p-value, but that data is not presented within this manuscript.

Supplemental Figure 6 in the text is said to describe a TCGA set of Stage III melanoma, but it appears the curves being described are in Supplementary Figure 5 but are labeled as Stage IV melanoma. (Many of the Supplemental Figures have been misnumbered without sufficient explanation in figure legends.)

It would have been intriguing to see an unsupervised clustering of the dataset in Figure 4D, rather than the separation as shown. Though some genes did reach $P < 0.05$, the heterogeneous nature of the visual provided and the number of genes evaluated likely require a correction for number of looks and may not be significant.

One has to be careful of associations with predicted neoantigen load, as the prediction algorithms are much more robust for some HLA alleles than they are for others and (while improving) less reliable for

Class II restricted predictions.

The loss of B2M in some of these tumors leads to interesting results, and the authors have chosen to discuss the patients that supported their hypothesis. Worthy of discussion are the three patients with B2M loss who experienced clinical benefit. It cannot be directly ascertained from Figure 3 if these patients were objective responders. (As a side note, the color schema for Figure 3 is confusing, as the same color can denote different things depending on the row in which it appears). Supplemental Figure 3 needs reformatting and does not add to the manuscript in its current form.

Speaking to these findings as potential biomarkers for choosing patients to treat with TIL lends a conundrum that the authors should address. These analyses have to be performed after a patient has undergone a surgical excision for TIL, therefore the patient has already undergone what is likely an operation not clinically indicated for any other reason, i.e. exposing them to risk for an experimental therapy. Without clear non-overlapping predictors, patients that could benefit might be excluded from further treatment. Additionally, waiting for analyses such as these could delay TIL transfer.

There is incredible potential within this dataset and the bioinformatics that have been used to analyze it, but the manuscript could be improved if it addressed the limitations of the small sample size, particularly when application to a larger data set (TCGA) does not always recapitulate the results.

Reviewer #2 (Remarks to the Author):

SUMMARY

The authors describe a comprehensive genomic analysis of responders and non responders to adoptive cell transfer therapy as a second line of treatment of melanoma. They find the mutational load, the predicted neoepitope load, and up-regulated MHC class I related gene signatures to be correlated with response.

The manuscript is very well written, the objective is clear, and the results are clearly presented. A pleasure to read!

MAJOR COMMENTS

What degree of correlation do you observe between mutational load and predicted neoantigen load? In any random sample, one expects to find roughly 1 predicted HLA binder for every 3 nonsynonymous mutations, and your Figures 2A+B greatly resemble Figures 5A+B.

To what degree do you observe correlation between mutational load and elevated activation of immune cells in the tumor micro environment and/or up-regulated MHC class I associated genes? It could be speculated that TIL homing is dictated by the presence of epitopes in the tumor.

In your GSEA, did you observe enrichment of homologous recombination genes or other DNA repair genes between responders and non responders? This could either be transcriptional or mutation aberrations (one approach for the latter is predicting the functional impact of mutations using e.g. PROVEAN). The reason I am asking is that Hugo et al. observed correlation between BRCA2 mutated genes and mutational load, which makes a lot of sense.

The four questions above basically elude to this: Could be that a more causative response predictor exists? It seems logical that response is driven by increased neoepitope load (shown by multiple other

studies), which is driven by increased overall mutational load (by pure random chance), which is driven (among other things) by dysfunctional DNA repair, which might be elucidated by a causative mutational or transcriptional profile (also shown in multiple other studies largely focussing on the BRCA genes)? This might be a reach, but it should be relatively easy to test using your comprehensive data.

Page 12: Section titled "Neoantigens in melanoma tumors predict benefit from ACT"

I think this is the major contribution from this study, but how would one go about predicting benefit from ACT using these data? Although the difference in predicted neoantigen load between responders and non responders is statistically significant, there is a large overlap in the number of predicted neoepitopes in the two groups (Figure 5B). Could you produce a predictive model if you included all your explanatory variables (insofar as they do not all correlate)? What is the accuracy of such a model in cross-validation, or, better yet, can it be validated on an external cohort if such exists?

You write that you have "pre-treatment" tumor sample - is this pre-ACT treatment, or pre-systemic therapy? If this is pre-systemic treatment, one would expect greater similarity to the Hugo et al. cohort. If it is pre-ACT, but post-systemic treatment, one would expect your cohort (all systemic treatment non-responders) to bear greater resemblance to the non-responders in the Hugo cohort. Such a comparison might be more useful than a comparison to the entire Hugo cohort consisting of both responders and non-responders.

MINOR COMMENTS

Do you observe Any correlation between response/survival and previous treatment? In any case, it would be nice to add a "previous treatments" bar to the bottom of Figure 1.

The term "neoantigen" is used rather liberally throughout the manuscript. I would recommend distinguishing between predicted HLA binders and bonafide T cell epitopes. Van Rooij et al. 2013 reported that roughly 2 out of every 500 predicted HLA binders were actually neoepitopes in melanomas. Obviously the more binders are predicted, the more like one will be to find a bonafide T cell epitope, but HLA binding (although believed to be rate limiting) is not the only factor determining T cell response.

Have you considered using, for example, DEseq2 or EdgeR to estimate differential expression and fold change? These widely used tools allow for factoring in more complex experimental designs (i.e. you could include disease type, previous treatment, etc. in your model) and enables linear modelling of survival.

Are you planning to release your analysis scripts along with your data?

Reviewer #3 (Remarks to the Author):

Review of Harbst et al..

Mutational and neoantigen load predict clinical benefit of adoptive T cell therapy in melanoma

In this paper, the authors attempt to find predictive biomarkers for clinical benefit from adoptive T cell therapy. They sequenced the exomes and transcriptomes of 27 patients that have received adoptive T cell therapy. They found that mutational load and neoantigen load predicted good response from ACT. This is a very nice manuscript that describes some very important observations. While mutational load

has been shown to predict clinical benefit from immune checkpoint therapy, no widely applicable biomarker has been found for ACT. This study begins to describe the tumor features that associate with good ACT response. All in all, this is important data that should be made available to the cancer research field.

Comments:

1. The authors should be more clear in whether deletion in the interferon signaling genes (as published by the Sharma lab in Cell associates with resistance. The IFN locus is on 9p, right next to CDKN2A. Do deletions near CDKN2A impact the IFN genes and do they track with response?
2. the author should clearly describe the outcomes of patients with B2M deletions. Were these single allele loss of homozygous deletions? Did some patients respond (n=2) and if so, how can this be explained?
3. was the IPRES signature described by Hugo et al Cell predictive of response in this cohort? The authors should show this data.

Reviewer #1 (Remarks to the Author):

Mutational and neoantigen load predict clinical benefit of adoptive T cell therapy in melanoma

The major claims of this manuscript are that a high rate of non-synonymous mutations and higher predicted neoantigen load are associated with clinical benefit in patients with metastatic melanoma treated with adoptive transfer with TIL. In addition, a strong immune activation signature is also associated with clinical benefit.

Given the current need for predictive biomarkers for immunotherapy of all kinds, the research questions addressed in this manuscript are vital to ongoing clinical efforts of adoptive cell transfer.

Points to be addressed:

It is an ongoing challenge in this field that “predicted neoantigens” and “neoantigens” continue to be used interchangeably, with few studies experimentally validating in silico predictions with in vitro recognition.

Reply: We agree with the reviewer and have clarified the potential nature of the neoantigens predicted in the study throughout the text of the manuscript.

When defining clinical benefit as a means of grouping patients, it is necessary to know if the patients received any therapy after adoptive cell transfer. It would also be of interest to know why a time period of 2 years was selected as a cut-off, as the majority of studies that have begun to use this criteria (checkpoint blockade) often use 6 months. Additionally, from the overall survival curves provided (Supp Fig 1), it appears that perhaps 3 or 4 responding patients did not survive 24 months (three steps and a censor mark?) which doesn't account for what may be biological differences between a patient with a >2y stable disease and a patient with a short-lived PR with limited survival.

Reply: We thank the reviewer for indicating this and giving us the opportunity to make this clearer in the manuscript. In the cohort of 27 patients enrolled in the clinical phase I/II trial of ACT the median overall survival was 22.8 months, hence the selection of 24 months as a cut-off. Would we have chosen a cut-off of 6 months OS, all but 6 patients would have passed this threshold. This would have given us less statistical power to conduct the analyses presented in the manuscript.

As the reviewer correctly indicates, from the data shown in the previous version of Supplementary Figure 1, it might appear that three patients with a partial response according to RECIST 1.0 did not survive > 24 months. However, when we generated that figure for the first submission of the manuscript, the minimum follow up was 22.6 months, and one RECIST responder had a follow up of exactly 22.6 months. Now, with longer follow up (updated with new database lock on June 27th 2017), this patient is still alive 37.2 months after TIL infusion. Thus, only two patients with RECIST responses survived less than 24 months after TIL infusion, notably 20.1 and 21.8 months – which is very close to the chosen cut off of 24 months that we used to define “clinical benefit” for patients without confirmed RECIST responses. As per clinical protocol and RECIST guidelines, all clinical responses in this trial are confirmed with one additional scan performed at least 6 weeks, after the first scan where >30% tumor regression is observed.

Therefore, we believe that all patients with confirmed RECIST responses should be included in the “clinical benefit”, regardless of survival time (that, in both cases, it is very close to the cut-off of 24 months), because tumor regression of chemotherapy-refractory malignancies such as melanoma is expected to have an important biological significance. Thus, we believe that we must take into account the effects of treatment on tumor masses. We have added some text on the definition of clinical benefit and no clinical benefit on page 4-5 under the section “Patient cohort and material”.

As the reviewer suggested, we have now included information on subsequent therapies, and when these were initiated, in Table 1.

In addition, we have updated all survival analyses with overall and progression-free survival locked on 27th of June, 2017. Hence, with the increased follow-up period there were minor revisions of the p-values from survival analyses that, importantly, did not change the significance of the findings. The p-values in Figures 2C and 2D changed from 8×10^{-4} and 0.02 to 8×10^{-4} and 0.007, respectively; the p-values in Figure 4C changed from 0.2 and 0.008 to 0.1 and 0.003, respectively; the p-values in Figures 5 C and 5D changed from 0.007 and 0.001 to 0.007 and 0.002, respectively.

The patients were stratified into three groups for mutational burden, but it is unclear how those groups were chosen. Was it driven by median values of the experimental group? Split into three equally populated groups? Or binned by discretely chosen values? It would be helpful to know more not only about the grouping, but about the filtering criteria applied to the samples in defining non-synonymous mutations. If this assessment of mutational burden is to be validated by other groups as a potential biomarker, it needs to be reproducible.

Reply: We thank the reviewer for raising an important issue. Importantly, we agree with the reviewer if mutational burden is to be used as a predictive test it needs to be robust and explicitly described. In this exploratory study, we divided all analysed patients into three equally sized groups (tertiles, n=8). The Low group had a median mutational burden of 98 mutations (range 23-193), the Intermediate group 286 mutations (range 194-495) and the High group 647 mutations (range 496-2200). We have also added the same information for the neoantigen grouping - Regarding expressed predicted neoantigens, the thresholds of the equal sized groups were: Low, 4 to 18; Intermediate, 19 to 56; High, 57 to 183.

Moreover, we have indicated more explicitly the median mutational load in each of the three groups on page 10 (row 225-227) as well the details of expressed neoantigens on page 15 (row 367-369). We also realize that we have not reported the definition of mutational burden in sufficient detail. We applied whole-exome sequencing using the Illumina sequencing platform on Agilent SureSelect targeted regions (median coverage 104x) and aligned the reads to the human genome hg19 using Novoalign. We kept single and double nucleotide mutations that were detected with both Mutect and VarScan softwares as described in Harbst et al ¹. Indels were not considered. The mutations were annotated using AnnoVar software (Table 1.1). We have now made this clearer in the manuscript in the methods section on page 5-6 under the section “WES data analysis”.

Table 1.1. Classification of mutation calls

Missense_Mutation	7021
Silent	3917

Nonsense_Mutation	432
Splice_Site	307
Intron	166
RNA	71
IGR	34
3'UTR	17
5'UTR	16
Start_Codon_SNP	11
lincRNA	5
Nonstop_Mutation	4

Moreover, we calculated the mutational burden as the sum of all mutations, including synonymous mutations. We now report the full procedure to obtain mutational burden and the respective grouping thresholds. Instances where we erroneously have indicated “non-synonymous” mutational burden, were corrected.

The present study reports a remarkable effect size of mutational burden in Kaplan-Meier analyses; however on a limited study cohort. In the revised manuscript, we also stress that the finding warrants a study on a larger cohort to refine the thresholds of mutational burden and to determine the clinical value of the proposed ACT-predictive markers. This has been added in the discussion section on page 17 (row 406-418).

The expression data is intriguing and nicely presented but would be enhanced by an explanation of the derivation of the MHC gene-expression score. Also, the TCGA and Cirenajwis cohorts were described as a validation set for this score, with a significant p-value, but that data is not presented within this manuscript.

Reply: We thank the reviewer for this. The genes involved in MHC-I antigen-processing and presentation displayed high correlation of gene expression (Figure 1.1). In particular, HLA-A, HLA-B,

HLA-C, TAP1, TAP2, NLRC5, PSMB9, PSMB8, and B2M were highly correlated, and further termed the “core” MHC-I set (boxed-in genes, Figure 1.1). The mean expression of these core genes is the MHC-I score and was then divided in quartiles to test the association with overall survival. For the MHC-I score quartiles, the resulting bins were: 4th quartile [-2.98,-0.493], 3rd quartile (-0.493,0.196], 2nd quartile (0.196,0.728], 1st quartile (0.728,2.16]. Details of the MHC-I score have been added to the Methods section on page 7 under the section “Expression signatures” of the revised manuscript as well as Supplementary Figure 1. This analysis shows in principle that a high MHC-I score is favorable for survival in our ACT treated cohort; but the finding requires validation in larger ACT studies.

Furthermore, we have included the MHC-I score analysis in the Cirenajwis cohort² and TCGA data as Supplementary Figure 6 of the revised manuscript (Figure 1.2). To stratify the patients we again used the MHC-I quartiles determined in the respective datasets. The complex effector T-cell mechanisms that occur, over time, in these untreated tumors may resemble the ACT approach that enforces a rapid multi-targeted T-cell response. Thus, the MHC-I score also has prognostic effects in melanoma.

Supplemental Figure 6 in the text is said to describe a TCGA set of Stage III melanoma, but it appears the curves being described are in Supplementary Figure 5 but are labeled as Stage IV melanoma. (Many of the Supplemental Figures have been misnumbered without sufficient explanation in figure legends.)

Reply: We thank the reviewer for this and we have thoroughly gone through all the Supplementary Figures and their legends. Unfortunately, there was a mistake in the text; it is meant to be stage IV. This has now been corrected.

It would have been intriguing to see an unsupervised clustering of the dataset in Figure 4D, rather than the separation as shown. Though some genes did reach $P < 0.05$, the heterogeneous nature of the visual provided and the number of genes evaluated likely require a correction for number of looks and may not be significant.

Reply: An unsupervised analysis of the gene expression data, using the most varying genes, largely recapitulates the intrinsic expression phenotypes that we proposed previously in Jönsson et al³

(Figure 1.3 left panel). Basically, the patients are divided into the “High-Immune” and “MITF-high Pigmentation” groups as only four tumors were classified as “MITF-low Proliferative”. These gene expression phenotypes are not sufficiently associated with clinical response. Also, the single immune cell makers do not separate patients with or without clinical benefit in unsupervised analysis (Figure 1.3, right panel). Nevertheless, an existing immune process, in particular represented by MHC-I, MHC-II and Interferon related gene sets, was significantly associated in GO-term and GSEA analysis. We therefore had picked the key genes of the respective immune pathways to visualize this effect in Figure 4D. Single genes of these pathways generally do not significantly discriminate between patients with or without clinical benefit, and the reviewer rightfully suggests correction for multiple testing. However, the purpose of Figure 4D was to visualize the key genes of these significant immune pathways. This comes also with the aim to give a fair report of the heterogeneity in these pathways across response groups, which may complicate the clinical application of these immune markers. In the light of these results we have not added the unsupervised analysis in Figure 4, however we have added the unsupervised analysis with the 1,500 most variable genes as Supplementary Figure 4 and a sentence on this on page 12 (row 282-284).

One has to be careful of associations with predicted neoantigen load, as the prediction algorithms are much more robust for some HLA alleles than they are for others and (while improving) less reliable for Class II restricted predictions.

Reply: We agree with the reviewer in that there is room for improvement within the neoantigen prediction field, and especially so for MHC class II predictors. Therefore, we have not used MHC class II predictions in our study. For MHC class I, we have used NetMHC; this algorithm has shown superior performance in benchmarking studies^{4, 5}. Even though these studies include older versions of NetMHC, while the most current benchmarking of available tools in Trolle et al⁶ unfortunately does not include NetMHC, it remains one of the most widely used in the field. Furthermore, while NetMHC v3.4 was implemented in the pVAC-Seq version available at the time the analyses were performed, we chose to implement NetMHC v4.0 just released at the time, with substantial improvements for prediction of longer binders. We have also included expression at nucleotide base level to further restrict predicted neoantigens to only expressed ones, adding an independent level of validation (RNA-seq level). Taking all of this together, we believe that neoantigens were derived in an optimal way, given the resources and the scope of the study. We have made this clearer in the methods section on page 7 (row 151-152).

The loss of B2M in some of these tumors leads to interesting results, and the authors have chosen to discuss the patients that supported their hypothesis. Worthy of discussion are the three patients with B2M loss who experienced clinical benefit. It cannot be directly ascertained from Figure 3 if these patients were objective

responders. (As a side note, the color schema for Figure 3 is confusing, as the same color can denote different things depending on the row in which it appears). Supplemental Figure 3 needs reformatting and does not add to the manuscript in its current form.

Reply: We thank the reviewer for allowing us to elaborate more on this. In total, there were 6 cases with B2M loss. 3 of these belonged to the clinical benefit group and 3 belonged to the group with no clinical benefit. Notably, 2 of the 3 cases belonging to the group with no clinical benefit had a progressive disease, thus being true objective non responders. The third case among the patients with no clinical benefit had a stable disease. In contrast, 2 of the cases in the clinical

benefit group were both true objective responders according to RECIST 1.0 as they displayed partial response. The third case had a stable disease with a PFS of 5.8 months but an overall survival > 24 months. Importantly, none of the analysed tumors harbored a B2M somatic mutation suggesting that one functional allele is present. Indeed, the RNAseq data does not indicate any differences between tumors harboring deletion of B2M and those that do not show any evidence of deletion of the B2M locus (P=0.3, Figure 1.4).

We have now added a section on page 11 (row 265-269) and also added a colour bar to Figure 3 allowing direct comparison between gene changes and objective response according to RECIST 1.0. The colour scheme has also been revised to avoid confusion. We have also removed Supplementary Figure 3 as this does not add any information.

Speaking to these finding as potential biomarkers for choosing patients to treat with TIL lends a conundrum that the authors should address. These analyses have to be performed after a patient has undergone a surgical excision for TIL, therefore the patient has already undergone what is likely an operation not clinically indicated for any other reason, i.e. exposing them to risk for an experimental therapy. Without clear non-overlapping predictors, patients that could benefit might be excluded from further treatment. Additionally, waiting for analyses such as these could delay TIL transfer.

Reply: We agree with the reviewer that, without clear non-overlapping predictors, patients that could benefit might be excluded from further treatment. Indeed, so far several predictive biomarkers for checkpoint inhibitor immunotherapy have been proposed.

Importantly, patients undergoing checkpoint inhibitor therapy should not necessarily undergo any type of surgical or biopsy procedure before treatment (while TIL patients should anyway), unless indicated for other reasons.

Interestingly, several biomarkers described to date including mutational load in melanoma ^{4,5} and lung cancer ⁶ or PD-L1 expression in several tumor types are not clear non-overlapping predictors ⁷, meaning that patients with a negative biomarker can still obtain clinical benefit. In addition, these biomarkers have been initially studied in clinical trials on the basis of one additional biopsy not indicated for any other reason.

To the best of our knowledge, the only non-overlapping predictive biomarker is the MSI-H or dMMR (mismatch repair deficiency) tumor status in colon cancer ⁸. Hence, the only biomarkers so far approved for clinical use that can be used for patient selection are PD-L1 staining with IHC for patients with untreated lung cancer ⁹ and MSI-H or dMMR tumor status for any tumor type (<https://www.fda.gov/Drugs/InformationOnDrugs/ApprovedDrugs/ucm560040.htm?platform=hootsuite>). In the latter case, according to the pivotal data for the approval, for the majority of patients the MSI-H or dMMR tumor status was determined prospectively with IHC tests for dMMR or laboratory-developed, investigational polymerase chain reaction (PCR) tests for MSI-H status. Thus, MSI-H status for the remaining patients (n=14) was determined through a retrospective evaluation of 415 tumor samples using a central laboratory-developed PCR test.

In our manuscript, we show that high mutational load in a tumor obtained with a surgical biopsy to be used for TIL manufacturing, is associated with better outcome after TIL therapy. This discovery strategy matches previous trials e.g. the MSI-H or dMMR, where prospective evaluation was used initially. Later in clinical development, retrospective evaluation was used. A similar strategy of clinical development could be envisaged for our biomarker, namely mutational load. Hence, genomic analyses should not necessarily be performed on the lesion removed for T-cell manufacturing but on a biopsy taken earlier in the metastatic disease/treatment course before enrolment in a TIL therapy protocol, but knowing that TIL therapy could be a potential treatment later on. Because of its technical complexity and toxicity, TIL therapy is not a 1. line therapy in melanoma and it is mostly intended as a “salvage” treatment”. Thus, there should be plenty of time to perform the analyses while patients are on-treatment with other agents and/or during TIL manufacturing, which typically takes 4 weeks or more. We have added a section on this in the discussion on page 17 (row 406-418).

There is incredible potential within this dataset and the bioinformatics that have been used to analyze it, but the manuscript could be improved if it addressed the limitations of the small sample size, particularly when application to a larger data set (TCGA) does not always recapitulate the results.

Reply: We have now stressed in the manuscript that our results need further validation in larger cohorts.

Reviewer #2 (Remarks to the Author):

SUMMARY

The authors describe a comprehensive genomic analysis of responders and non responders to adoptive cell transfer therapy as a second line of treatment of melanoma. They find the mutational load, the predicted neoepitope load, and up-regulated MHC class I related gene signatures to be correlated with response.

The manuscript is very well written, the objective is clear, and the results are clearly presented. A pleasure to read!

MAJOR COMMENTS

What degree of correlation do you observe between mutational load and predicted neoantigen load? In any random sample, one expects to find roughly 1 predicted HLA binder for every 3 nonsynonymous mutations, and your Figures 2A+B greatly resemble Figures 5A+B.

Reply: There is indeed a high Pearson correlation of 0.98 between mutational load and the number of expressed predicted neoantigens (Figure 2.1). In our data, there is one expressed predicted neoantigen for every 6.5 non-synonymous mutations, which is in line with the estimate of the reviewer, who disregarded the expression status of the neoantigens. The correlation value is now mentioned in the revised manuscript on page 15 (row 371).

To what degree do you observe correlation between mutational load and elevated activation of immune cells in the tumor micro environment and/or up-regulated MHC class I associated

genes? It could be speculated that TIL homing is dictated by the presence of epitopes in the tumor.

Reply: Mutational load did not correlate to signatures of melanoma-derived T-cells, B-cells and macrophages (Tirosch et al., Science 2016) (Figure 2.2, upper panel). Also, mutational load did not correlate with the MHC-I score (Figure 2.2, lower panel). The same results were observed for predicted neoantigen load and log10-transformed mutational load. These results have now been added to the results section on page 14 (row 340-345) Figure 2.2 has also been added as Supplementary Figure 10.

To systematically determine the associations of mutational load to gene expression of bulk tumors, we used the TCGA samples from lymph node metastases and extracted the lower and upper sample quintiles in respect to number of non-silent mutations (lower quintile 2-113 mutations, $n=45$; upper quintile 773-16,000 mutations, $n=44$). SAM-analyses yielded only 27 genes to be up-regulated in the highly mutated samples ($fdr < 0.05$) compared to the least mutated samples. These genes did not display enrichment for GO terms. Initial reports proposed the hypothesis that a high mutational load would attract T-cells to the tumor site. However, there is emerging evidence that such a correlation is absent at least in melanoma^{10,11}. We did not find a correlation of immune cell infiltration and mutational load in our heavily pre-treated cohort and in largely untreated regional lymph node metastases from TCGA.

In your GSEA, did you observe enrichment of homologous recombination genes or other DNA repair genes between responders and non responders? This could either be transcriptional or mutation aberrations (one approach for the latter is predicting the functional impact of mutations using e.g. PROVEAN). The reason I am asking is that Hugo et al. observed correlation between BRCA2 mutated genes and mutational load, which makes a lot of sense.

Reply: The gene BRCA2 displayed a missense mutation in three patients, 1 belonging to the group with no clinical benefit and 2 in the clinical benefit group. Also, SERPINB3 and SERPINB4 genes that have been found mutated in CTLA4i responders¹², were not differentially mutated in our cohort (1

without clinical benefit, 3 with clinical benefit). We then systematically investigated which genes were differentially mutated between patients with or without clinical benefit. To avoid an over-representation of large genes affected by random mutations in samples with high mutational burden, we performed logistic regression with adjustment for mutational burden. No single gene reached an $fdr < 0.05$. We then used GO term analysis for the 150 genes with the lowest p-values for over-mutation in patients with clinical benefit, which resulted in enrichment of developmental terms ("heart development", $fdr = 0.006$), but not enrichment of DNA-repair associated terms. (For no clinical benefit, over-mutated genes were not significantly enriched for any GO term.) Re-doing the analysis without adjustment for mutational burden, but instead taking the genes with at least three excess mutations in the clinical benefit group ($n=100$ genes), also did not yield DNA-repair related GO terms.

We then re-evaluated the transcriptional GO term and GSEA analyses in regard to DNA-repair signatures. GO-term analysis for neither responder-enriched genes nor non-responder-enriched genes yielded DNA repair related terms. For GSEA, there were no DNA-repair related processes enriched in responders. However, for the non-responders, we observed enrichment of the pathways "REACTOME_CHROMOSOME_MAINTENANCE" ($fdr=0$) and "REACTOME_MEIOTIC_RECOMBINATION" ($fdr=0$). However, the GSEA list was dominated by a large amount of cell-cycle and proliferation related signatures (Supplementary Table 1 of manuscript). We would therefore argue that the elevated DNA-repair signatures in non-responders are due to a correlation with a more proliferative cell state, rather than implying a down-regulation of DNA repair genes in responders.

Collectively, with this sample size, there is insufficient evidence that a dysfunctional DNA repair has caused a higher mutation rate in responders. This is also in agreement to the finding that the vast majority of cutaneous melanomas in our ACT cohort, as well as in a large prospectively collected Australian cohort¹³ are C>T mutations in a context that is typical for UV-induced mutations, i.e. Mutational Signature 7¹⁴.

The four questions above basically eludes to this: Could be that a more causative response predictor exists? It seems logical that response is driven by increased neoepitope load (shown by multiple other studies), which is driven by increased overall mutational load (by pure random chance), which is driven (among other things) by dysfunctional DNA repair, which might be elucidated by a causative mutational or transcriptional profile (also shown in multiple other studies largely focussing on the BRCA genes)? This might be a reach, but it should be relatively easy to test using your comprehensive data.

Reply: See response above. We believe that this is not sufficient evidence to claim that DNA repair mechanisms are responsible for poor response to ACT. We have indicated the BRCA2 mutations in Figure 3A and on page 10 (row 245-248) of the revised manuscript and reported that GO term and GSEA analyses did not find DNA repair pathways to be associated to response at the mutational nor transcriptional level on page 13 (row 299-300).

Page 12: Section titled "Neoantigens in melanoma tumors predict benefit from ACT"

I think this is the major contribution from this study, but how would one go about predicting benefit from ACT using these data? Although the difference in predicted neoantigen load between responders and non responders is statistically significant, there is a large overlap in the number of predicted neoepitopes in the two groups (Figure 5B). Could you produce a predictive model if you included all your explanatory variables (insofar as they do not all correlate)? What

is the accuracy of such a model in cross-validation, or, better yet, can it be validated on an external cohort if such exists?

Reply: To make the analysis reproducible, we mentioned the thresholds of expressed predicted neoantigens for the equal-sized groups in the revised manuscript: low, 4 to 18; intermediate, 19 to 56; high, 57 to 183.

In order to obtain a fair estimate of the accuracy in prediction of response, we used logistic regression to model response using the logged predicted neoantigens and optionally including the (uncorrelated) MHC-I score. The accuracy was then determined in 100 repeats of a 5x cross-validation loop. (In each loop, using ROC analysis to determine the optimal cutoff for the fitted values in the training set, and applying the cutoff to the test set). The mean accuracy in cross validation for the log10-transformed neoantigen count was 0.63, and the mean accuracy of the model with log10-transformed neoantigen count plus MHC-I score was 0.68. Alternatively, in multivariate Cox regression, log10-transformed predicted neoantigen count ($p=0.007$) and MHC-score ($p=0.01$) were independent predictors of overall survival. To date, there is no external ACT data available for validation.

While these results are interesting, we think that the present cohort is too small to propose a fixed predictor algorithm for clinical application. It is also possible that ACT may show different effects on immediate response (RECIST) and on long-term survival of patients. We argue in the revised manuscript that a larger cohort will be needed to determine a fully specified predictor of patient outcome. However, we have added the results of the multivariate analysis using log10-transformed predicted neoantigen count and MHC-score as variables on page 16 (row 374-379). A sentence in the discussion section page 19 (row 456-459) has also been added.

You write that you have “pre-treatment” tumor sample - is this pre-ACT treatment, or pre-systemic therapy? If this is pre-systemic treatment, one would expect greater similarity to the Hugo et al. cohort. If it is pre-ACT, but post-systemic treatment, one would expect your cohort (all systemic treatment non-responders) to bear greater resemblance to the non-responders in the Hugo cohort. Such a comparison might be more useful than a comparison to the entire Hugo cohort consisting of both responders and non-responders.

Reply: All analysed tumor samples included in the manuscript are pre-ACT treatment and not pre-systemic treatment samples. Hence, these tumors are surgically removed after progression or while on prior immunotherapy. We agree that the Hugo cohort is not the best suitable test cohort and thus we have initially toned down the results obtained in the Hugo cohort.

MINOR COMMENTS

Do you observe Any correlation between response/survival and previous treatment? In any case, it would be nice to add a “previous treatments” bar to the bottom of Figure 1.

Reply: We have now added a bar representing previous treatment in Figure 1.

The term “neoantigen” is used rather liberally throughout the manuscript. I would recommend distinguishing between predicted HLA binders and bonafide T cell epitopes. Van Rooij et al. 2013 reported that roughly 2 out of every 500 predicted HLA binders were actually neoepitopes in melanomas. Obviously the more binders are predicted, the more like one will be to find a bonafide T cell epitope, but HLA binding

(although believed to be rate limiting) is not the only factor determining T cell response.

Reply: We have throughout the manuscript revised the usage of “neoantigen”.

Have you considered using, for example, DEseq2 or EdgeR to estimate differential expression and fold change? These widely used tools allow for factoring in more complex experimental designs (i.e. you could include disease type, previous treatment, etc. in your model) and enables linear modelling of survival.

Reply: For this manuscript we have used FPKM values that were obtained from the Tophat-Cufflinks pipeline. FPKM values are normalized for exonic length and library size and the log-transformed FPKM-values have a mean-variance relationship that is comparable to microarray data. Therefore, statistical methods that have been settled in the microarray field can be applied, including linear models and survival analyses. In addition, the normalization step ensures that the genes as well as samples are directly comparable with each other, without the use of scaling factors or adjustment for e.g., library size. We are aware that modeling raw counts for differential expression under the assumption of a negative binomial distribution (DEseq2, EdgeR) has become a popular alternative to a fully normalized dataset. For the ease of data comparability and visualization purposes we have, however, decided to use FPKM values and classical statistics.

Are you planning to release your analysis scripts along with your data?

Reply: We are releasing the processed expression data at Gene Expression Omnibus as GSE100797, and segmented copy number data, mutation data and clinical annotations as Supplementary Data, to give other researchers the possibility to comprehensively analyze the data. Also, we have added the grouping thresholds to the revised manuscript, so that the findings can be reproduced by others. As we applied standard statistical methods, we would argue that the scripts are not essential.

Reviewer #3 (Remarks to the Author):

Review of Harbst et al..

Mutational and neoantigen load predict clinical benefit of adoptive T cell therapy in melanoma

In this paper, the authors attempt to find predictive biomarkers for clinical benefit from adoptive T cell therapy. They sequenced the exomes and transcriptomes of 27 patients that have received adoptive T cell therapy. They found that mutational load and neoantigen load predicted good response from ACT. This is a very nice manuscript that describes some very important observations. While mutational load has been shown to predict clinical benefit from immune checkpoint therapy, no widely applicable biomarker has been found for ACT. This study begins to describe the tumor features that associate with good ACT response. All in all, this is important data that should be made available to the cancer research field.

Comments:

Figure 3.1. Amplifications and losses of Interferon gamma-related genes and *CDKN2A*. Interferon gamma genes were defined as described by Gao et al., Cell 2016. Deletions (blue) were called with log ratios < (-0.5) and amplifications (red) with log ratios > 0.5, as in Gao et al., Cell 2016.

1. The authors should be more clear in whether deletion in the interferon signaling genes (as published by the Sharma lab in Cell associates with resistance. The IFN locus is on 9p, right next to *CDKN2A*. Do deletions near *CDKN2A* impact the IFN genes and do they track with response?

Reply: Deletions in Interferon-related genes, as defined in the Cell publication from the Sharma lab¹⁵, are relatively frequent in melanoma and occur preferably at chromosome 9 (Figure 3.1). Indeed, these interferon gene deletions on chromosome 9 co-occur with CDKN2A deletions; albeit there are three additional samples which harbor an exclusive CDKN2A deletion. None of the deletions in the Interferon-related genes, nor CDKN2A, is associated with response ($p > 0.2$ for all comparisons). The Gao et al. publication¹⁵ has investigated treatment with the CTLA4i drug ipilimumab, while our cohort was treated with adoptive T-cell therapy. Nevertheless, we have added a sentence on deletions of the IFN locus on chromosome 9p on page 11 (row 256-261) in the manuscript.

2. the author should clearly describe the outcomes of patients with B2M deletions. Were these single allele loss of homozygous deletions? Did some patients respond (n=2) and if so, how can this be explained?

Reply: We thank the reviewer for allowing us to elaborate more on this. In total, there were 6 cases with B2M loss. 3 of these belonged to the clinical benefit group and 3 belonged to the group with no clinical benefit.

Figure 3.2. Boxplot of B2M gene expression values in comparison to B2M deletion or not.

Notably, 2 of the 3 cases belonging to the group with no clinical benefit had a progressive disease, thus being true objective non responders. The third case among the patients with no clinical benefit had a stable disease. In contrast, 2 of the cases in the clinical benefit group were both true objective responders according to RECIST 1.1 as these displayed partial response. The third case had a stable disease with a PFS of 5.8 months but an overall survival > 24 months. Importantly, none of the analysed tumors harbored a B2M somatic mutation

suggesting that one functional allele is present. Indeed, the RNAseq data does not indicate any differences between tumors harboring deletion of B2M and those that do not show any evidence of deletion of the B2M locus (Figure 3.2). A section on B2M deleted cases has been added on page 11 (row 265-269). We have now also added a colour bar to Figure 3 allowing direct comparison between gene changes and objective response according to RECIST 1.1.

3. was the IPRES signature described by Hugo et al Cell predictive of response in this cohort? The authors should show this data.

Reply: We have now applied the IPRES signatures to our cohort as described in Hugo et al.¹⁶. (21 of 22 validated IPRES signatures were available; mean z-score of signature GSVA scores; cutoff of 0.35 to be called “IPRES-enriched”). There were 5 IPRES-enriched samples among the non-responders and 5 IPRES-enriched samples among the responders (P=1) (Figure 3.2, left panel). Overall survival rates were comparable for IPRES-enriched and IPRES-not-enriched samples (p=0.8) (Figure 3.3, right panel). The samples from Hugo et al. were treated with a PD1 inhibitor blocking a single target molecule, whereas our cohort received treatment with T-cells that potentially have multiple ways to target the cancer cells. Therefore, response mechanisms might differ between the treatments. These results have been added to the manuscript on page 12 (row 284-290) and Supplementary Figure 5. Description of the classification procedure is now explained in the methods section on page 7-8 (row 171-178).

References:

1. Harbst K, et al. Multiregion Whole-Exome Sequencing Uncovers the Genetic Evolution and Mutational Heterogeneity of Early-Stage Metastatic Melanoma. *Cancer Res* **76**, 4765-4774 (2016).
2. Cirenajwis H, et al. Molecular stratification of metastatic melanoma using gene expression profiling: Prediction of survival outcome and benefit from molecular targeted therapy. *Oncotarget* **6**, 12297-12309 (2015).

3. Jonsson G, *et al.* Gene expression profiling-based identification of molecular subtypes in stage IV melanomas with different clinical outcome. *Clin Cancer Res* **16**, 3356-3367 (2010).
4. Van Allen EM, *et al.* Genomic correlates of response to CTLA-4 blockade in metastatic melanoma. *Science* **350**, 207-211 (2015).
5. Snyder A, *et al.* Genetic basis for clinical response to CTLA-4 blockade in melanoma. *N Engl J Med* **371**, 2189-2199 (2014).
6. Rizvi NA, *et al.* Cancer immunology. Mutational landscape determines sensitivity to PD-1 blockade in non-small cell lung cancer. *Science* **348**, 124-128 (2015).
7. Gandini S, Massi D, Mandala M. PD-L1 expression in cancer patients receiving anti PD-1/PD-L1 antibodies: A systematic review and meta-analysis. *Crit Rev Oncol Hematol* **100**, 88-98 (2016).
8. Le DT, *et al.* PD-1 Blockade in Tumors with Mismatch-Repair Deficiency. *N Engl J Med* **372**, 2509-2520 (2015).
9. Reck M, *et al.* Pembrolizumab versus Chemotherapy for PD-L1-Positive Non-Small-Cell Lung Cancer. *N Engl J Med* **375**, 1823-1833 (2016).
10. Spranger S, *et al.* Density of immunogenic antigens does not explain the presence or absence of the T-cell-inflamed tumor microenvironment in melanoma. *Proc Natl Acad Sci U S A* **113**, E7759-E7768 (2016).
11. Danilova L, *et al.* Association of PD-1/PD-L axis expression with cytolytic activity, mutational load, and prognosis in melanoma and other solid tumors. *Proc Natl Acad Sci U S A* **113**, E7769-E7777 (2016).
12. Riaz N, *et al.* Recurrent SERPINB3 and SERPINB4 mutations in patients who respond to anti-CTLA4 immunotherapy. *Nat Genet* **48**, 1327-1329 (2016).
13. Hayward NK, *et al.* Whole-genome landscapes of major melanoma subtypes. *Nature* **545**, 175-180 (2017).
14. Alexandrov LB, *et al.* Signatures of mutational processes in human cancer. *Nature* **500**, 415-421 (2013).
15. Gao J, *et al.* Loss of IFN-gamma Pathway Genes in Tumor Cells as a Mechanism of Resistance to Anti-CTLA-4 Therapy. *Cell* **167**, 397-404 e399 (2016).
16. Hugo W, *et al.* Genomic and Transcriptomic Features of Response to Anti-PD-1 Therapy in Metastatic Melanoma. *Cell* **168**, 542 (2017).

REVIEWERS' COMMENTS:

Reviewer #1 (Remarks to the Author):

The authors clearly addressed each of the questions raised in my previous review, improving the quality of the submitted manuscript.

My only additional question for consideration is regarding B2M deletion. While they have addressed the initial question well in the context of class I restricted recognition, I would hypothesize that the patients who experienced objective regression may have B2M-independent mechanisms of recognition.

Again, this is well-executed comprehensive analysis of a population and a potential salvage therapeutic option in need of further exploration. This manuscript is invaluable to those studying adoptive cell transfer.

Reviewer #2 (Remarks to the Author):

Thank you very much for an extremely thorough reply to my comments. At this point, I am satisfied with the changes made and the work is, in my opinion, of a quality ready for publication.

Reviewer #3 (Remarks to the Author):

The authors have addressed my comments. The paper reads very well.

Please find below reply to reviewer #1.

Reviewer #1 (Remarks to the Author):

The authors clearly addressed each of the questions raised in my previous review, improving the quality of the submitted manuscript.

My only additional question for consideration is regarding B2M deletion. While they have addressed the initial question well in the context of class I restricted recognition, I would hypothesize that the patients who experienced objective regression may have B2M-independent mechanisms of recognition.

Reply: We agree with the reviewer regarding the three patients with B2M deletions that had clinical response. In the Discussion section of revised manuscript, page 14, we have added that there may be B2M-independent mechanisms for T-cells to recognize the tumor.

We thank the reviewers for all comments improving the manuscript considerably.